

# Bi-Level ventilation decreases pulmonary shunt and modulates neuroinflammation in a cardiopulmonary resuscitation model

Robert Ruemmler[1], Alexander Ziebart[1], Frances Kuropka[1], Bastian Duenges[1], Jens Kamuf[1], Andreas Garcia-Bardon[1] and Erik K. Hartmann[1]

Department of Anesthesiology, Medical Centre of the Johannes Gutenberg University, Mainz, Germany

## ABSTRACT

**Background**. Optimal ventilation strategies during cardiopulmonary resuscitation are still heavily debated and poorly understood. So far, no convincing evidence could be presented in favour of outcome relevance and necessity of specific ventilation patterns. In recent years, alternative models to the guideline-based intermittent positive pressure ventilation (IPPV) have been proposed. In this randomized controlled trial, we evaluated a bi-level ventilation approach in a porcine model to assess possible physiological advantages for the pulmonary system as well as resulting changes in neuroinflammation compared to standard measures.

**Methods**. Sixteen male German landrace pigs were anesthetized and instrumented with arterial and venous catheters. Ventricular fibrillation was induced and the animals were left untreated and without ventilation for 4 minutes. After randomization, the animals were assigned to either the guideline-based group (IPPV, tidal volume 8–10 ml/kg, respiratory rate 10/min, $F_iO_2$ 1.0) or the bi-level group (inspiratory pressure levels 15–17 $cmH_2O$/5$cmH_2O$, respiratory rate 10/min, $F_iO_2$ 1.0). Mechanical chest compressions and interventional ventilation were initiated and after 5 minutes, blood samples, including ventilation/perfusion measurements via multiple inert gas elimination technique, were taken. After 8 minutes, advanced life support including adrenaline administration and defibrillations were started for up to 4 cycles. Animals achieving ROSC were monitored for 6 hours and lungs and brain tissue were harvested for further analyses.

**Results**. Five of the IPPV and four of the bi-level animals achieved ROSC. While there were no significant differences in gas exchange or hemodynamic values, bi-level treated animals showed less pulmonary shunt directly after ROSC and a tendency to lower inspiratory pressures during CPR. Additionally, cytokine expression of tumour necrosis factor alpha was significantly reduced in hippocampal tissue compared to IPPV animals.

**Conclusion**. Bi-level ventilation with a constant positive end expiratory pressure and pressure-controlled ventilation is not inferior in terms of oxygenation and decarboxylation when compared to guideline-based IPPV ventilation. Additionally, bi-level ventilation showed signs for a potentially ameliorated neurological outcome as well as less pulmonary shunt following experimental resuscitation. Given the restrictions of the animal model, these advantages should be further examined.

Corresponding author
Robert Ruemmler,
robert.ruemmler@email.de

## INTRODUCTION

Cardiac arrest and cardiopulmonary resuscitation (CPR) are regularly encountered scenarios in clinical, as well as pre-hospital situations. With incidence rates of up to 0.1% per year and an overall mortality rate close to 90%, there is still a lack of effective and evidence-based treatment options to prevent permanent damage or death (*Grasner et al., 2016*). While there have been changes to resuscitation guidelines in Europe and the US regarding early defibrillation and high-quality chest compressions, guideline-based ventilation recommendations have not been substantially changed in almost two decades (*Brooks et al., 2015*; *Callaway et al., 2015*; *Sandroni & Nolan, 2011*; *Tanaka et al., 2017*). In recent years, several alternatives to the standard intermittent positive pressure ventilation (IPPV) method with a fixed respiratory rate (RR) of 10 breaths per minute have been proposed. Ranging from synchronized ventilation in order to suppress chest compression interference (*Kill et al., 2015*) to ultra-low-tidal-volume ventilation to mitigate possible lung injuries (*Ruemmler et al., 2018*) up to mere passive oxygenation via high-flow oxygenation supply (*Deakin, O'Neill & Tabor, 2007*; *Koster et al., 2007*), many approaches have been tested and have shown—in parts—promising results.

However, due to the complexity of the topic and obvious ethical problems while designing and conducting prospective randomized-treatment resuscitation study protocols, convincing evidence to change current recommendations is still missing (*Rubulotta & Rubulotta, 2013*).

In this study, we wanted to further evaluate a previously proposed approach of a pressure-controlled bi-level ventilation model during cardiopulmonary resuscitation in swine (*Kill et al., 2014*). Based on our own research (*Ruemmler et al., 2018*), our main hypothesis was that this ventilation mode could result in lower tidal volumes and peak inspiratory pressures and might improve oxygenation parameters during and especially after return of spontaneous circulation (ROSC). Secondly, we assumed that these effects could provide favourable remote end organ effects and reduce hypoxic neuroinflammation after successful resuscitation.

## METHODS

### Anaesthesia/instrumentation

Following approval of the study by the State and Institutional Animal Care Committee Rhineland Palatine (Landesuntersuchungsamt Rheinland-Pfalz, approval no. G16-1-042), 16 male German landrace pigs (12–16 weeks, 30–35 kg) were acquired from a local farm and received pre-transport sedation via an intramuscular injection of azaperone and ketamine (4 mg/kg). Instrumentation and animal preparation as well as extended cardiovascular monitoring were carried out as described before by our group (*Ruemmler et al., 2018*).

Specifically, constant anesthesia was maintained during the entire experiment using propofol and fentanyl infusions, a base ventilation was established (6–8 ml/kgBW, PEEP 5 cmH$_2$O, peak inspiratory pressure of 40 cmH$_2$O, adapted respiratory rate to adequate CO$_2$ levels) and central venous and arterial access was established under ultra sound guidance (*Ruemmler et al., 2018*). A transpulmonary thermodilution catheter (PiCCO, Pulsion, Munich, Germany), a Swan-Ganz-catheter and an intravenous pacing catheter (Osypka Medical GmbH, Rheinfelden-Herten, Germany) were placed as established previously (*Ruemmler et al., 2018*). The fasting animals received an initial fluid bolus of 30 ml/kg balanced electrolyte solution and were left to stabilize for 30 min before baseline measurements were taken. Mean arterial blood pressure was maintained above a threshold of 50 mmHg using norepinephrine infusion, if necessary.

## Intervention

Following base line measurements, ventricular fibrillation was induced via the fibrillation catheter (13.8 V current at 200 Hz according to manufacturer's recommendation) and the ventilator was disconnected (*Ruemmler et al., 2018*). Monitor-confirmed cardiac arrest was permitted for four minutes and the animals were randomized into two groups by blinded drawing of one of 16 envelopes containing the respective ventilation mode (eight animals per group):

Group 1 ("IPPV", standard) )received guideline-based intermittent positive pressure ventilation (IPPV), V$_t$: 8–10 ml/kg, RR: 10/min, F$_i$O$_2$: 1.0

Group 2 ("bi-level") received bi-level ventilation with pressure levels of 15–17 cmH$_2$O maximum inspiratory pressure and 5 cmH$_2$O minimum pressure, RR: 10/min, F$_i$O$_2$: 1.0. Peak inspiratory pressure levels were determined by assessing the average pressure necessary to result in a pre-CPR tidal volume of 6–8 ml/kgBW.

Standardized mechanical chest compressions using a LUCAS-2 device (PhysioControl, Lund, Sweden) with a fixed compression rate of 100/min and the randomized ventilation mode were then initiated following an established protocol (*Ruemmler et al., 2018*). After 5 minutes of continuous CPR, blood samples were collected and resuscitation measures were continued according to the advanced life support algorithm: two minute compression cycles, rhythm analysis, defibrillation (200J, bi-phasic, electrodes in anterior-posterior position), epinephrine (1 mg) and vasopressine (0.1 U/kg) administration. If ROSC was not achieved after the 4th defibrillation, the experiment was terminated. Animals achieving ROSC were switched back to standard ventilation and monitored for six hours. During the monitoring period, mean arterial blood pressure was kept over 50 mmHg using a norepinephrine drip if necessary. The experiment was terminated with the animal being euthanized in deep anesthesia using high doses of propofol (200 mg) and potassium chloride (40 mmol).

## Measurements/sample collection

Cardiopulmonary and respiratory values were constantly measured and collected for the duration of the whole experiment as described before (*Ruemmler et al., 2018*) including spirometry, ventilation pressures and hemodynamics. Additionally, blood gas analyses

and cardiac output (CO) measurements were taken at baseline, during CPR, ten minutes post-CPR and hourly afterwards.

Ventilation/perfusion (V/Q) analyses were performed at baseline, during CPR and ten minutes post-CPR using the micropore membrane inlet mass spectrometry facilitated multiple inert gas elimination technique (MMIMS-MIGET, Oscillogy LLC, Philadelphia, USA) as described before (*Hartmann et al., 2014*). Specifically, subclinical, non-toxic doses of a saline solution containing six chemically inert gases with different elimination constants (sulphur hexafluoride, krypton, desflurane, enflurane, diethyl ether and acetone) were infused starting 20 min prior to measurements in order to reach an in vivo steady state. Blood samples from the pulmonary and femoral artery were taken and analyzed via a mass spectrometer determining gas elimination during lung passage, thus allowing accurate V/Q fraction determination for high, normal and low perfusion ratios as well as shunt volumes.

After termination, both lungs were harvested and samples from upper and lower left lung lobe were either snap frozen for biomolecular analyses or preserved in 2% formaldehyde solution for histologic fixation. The right upper lobe was used for wet-to-dry ratio measurements. A previously established tissue damage score was used in investigator-blinded manner to quantify histologic injury as described before (*Ziebart et al., 2015*; *Ziebart et al., 2014*).

RNA extraction and real-time polymerase chain reaction measurements were performed in lung tissue, cortex and hippocampus samples using a relative RNA quantification kit with a cyclophilin A (peptidylprolyl isomerase A, PPIA) reference and a Lightcycler 480 system (LightCycler, Roche, Mannheim, Germany) according to manufacturer's instructions in order to determine cytokine expression levels of proinflammatory interleukin 6 (IL-6) and tumor necrosis factor alpha (TNFα) (*Ruemmler et al., 2018*).

Statistical analyses were performed using 2-way ANOVA inter-group tests with post-hoc Bonferroni correction for repeated measurements (e.g., blood pressures, inspiratory pressures) as well as Mann–Whitney U test for single measurements (e.g., IL-6, TNFalpha, MIGET) with Student–Newman–Keuls post-hoc analysis via GraphPad Prism 8 software (GraphPad Software Inc., La Jolla, CA, USA). All data in the text are presented as mean (standard deviation). Bar plots are shown as mean with the standard error of the mean-$P$-values < 0.05 were considered significant.

## RESULTS

A total of 16 experiments were performed. ROSC was achieved in 5 of the IPPV and 4 of the bi-level animals (62.5% versus 50%). No substantial fractures, macroscopic injuries or pneumothoraces were detected in any of the animals after mechanical chest compressions. There were no significant differences in haemodynamic values between the two groups at any given time point (Table 1). There were neither inter-group differences in vasopressor need during and after successful CPR nor in the number of defibrillations to achieve ROSC. Horovitz's index ($PaO_2/FiO_2$) in surviving animals showed no difference over the whole monitoring period as well as during CPR (Fig. 1).

**Table 1  Overview of vital parameters.**

| Parameter<br>MEAN (SD) | | BLH | CPR | ROSC | 1 h | 5 h |
|---|---|---|---|---|---|---|
| HR | IPPV | 77(19) | 101(0) | 149(38) | 110(22) | 89(19) |
| [bpm] | bi-Level | 70(15) | 101(0) | 141(57) | 117(13) | 123(32) |
| MAP | IPPV | 75(7) | 34(17) | 105(13) | 72(10) | 72(9) |
| [mmHg] | bi-Level | 73(10) | 30(11) | 84(27) | 68(4) | 65(8) |
| CVP | IPPV | 10(2) | 29(12) | 12(1) | 11(3) | 9(4) |
| [mmHg] | bi-Level | 9(6) | 21(9) | 8(3) | 9(3) | 6(3) |
| PAP | IPPV | 22(4) | 52(25) | 22(9) | 25(2) | 19(3) |
| [mmHg] | bi-Level | 22(6) | 56(22) | 22(2) | 25(2) | 18(4) |
| CI | IPPV | 4.03(0.5) | # | 4.26(1.4) | 3.57(0.5) | 3.51(0.7) |
| [(l/min)/m$^2$] | bi-Level | 3.92(0.9) | # | 3.03(1.2) | 2.97(0.5) | 3.13(0.6) |
| NE | IPPV | 0 | 0 | 1.1(0.97) | 1(0.71) | 0.28(0.4) |
| [mg/h] | bi-Level | 0 | 0 | 0.6(0.43) | 1.25(0.92) | 0.13(0.15) |
| T | IPPV | 36.6(1) | 36.6(1) | 36.9(1.2) | 37.7(1.3) | 38.5(0.8) |
| [°C] | bi-Level | 36.4(0.8) | 36.5(1) | 36.8(0.3) | 37.1(0.7) | 38.4(0.9) |
| FRC | IPPV | 657(152) | # | 478(135) | 529(119) | 453(176) |
| [ml] | bi-Level | 676(102) | # | 480(123) | 517(210) | 471(146) |
| Lactate | IPPV | 2.1(1.1) | 6.1(2.3) | 9.3(2.5) | 6.8(1.8) | 1.9(0.7) |
| [mmol/l] | bi-Level | 1.9(1.4) | 5.8(2.3) | 7.3(3.4) | 8.2(0.9) | 2.5(0.7) |
| SvO$_2$ | IPPV | 69(19) | 28(8) | 51(19) | 48(6) | 51(5) |
| [%] | bi-Level | 67(9) | 25(5) | 51(13) | 51(3) | 52(14) |

Notes.

HR, heart rate; MAP, mean arterial pressure; CVP, central venous pressure; CI, cardiac index; PAP, pulmonary arterial pressure; NE, norepinephrine; T, T temperature; FRC, functional residual capacity; SvO$_2$, central venous oxygen saturation; BLH, base line healthy; CPR, measurement after 4 minutes CPR; ROSC, measurement 10 minutes post ROSC.

PaCO$_2$ values were not higher under bi-level ventilation during CPR and showed no difference after ROSC. Peak inspiratory pressures were significantly decreased during ROSC in the bilevel group ($P = 0.005$) but showed no difference in delivered tidal volumes (Fig. 2). Post ROSC, necessary inspiratory pressures tended to be decreased, but did not show statistical significance.

Ventilation/perfusion analyses showed no major differences in all lung areas (high V/Q, medium V/Q and low V/Q) during CPR but significantly less pulmonary shunt immediately post-ROSC in the bi-level group (IPPV: 26.8 (16.2) [% of cardiac output]; bi-level: 9.4 (8.2) [% of cardiac output], $P = 0.017$) (Fig. 3).

Lung histology showed no significant differences in tissue damage scores. The pulmonary wet-to-dry ratio did not differ between both groups. IL-6 and TNFα mRNA expression showed no difference in lung and cortical tissue but was decreased in hippocampal samples of the animals receiving bi-level ventilation with a statistically significant difference in TNFα expression (IPPV: 4.9 (2.3) [$10^{-5}$ pg/µl]; bi-level: 1.5 (.4) [$10^{-5}$ pg/µl], $P = 0.032$) (Fig. 4).

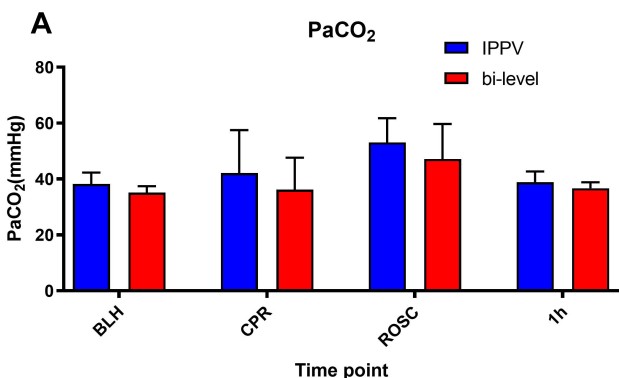

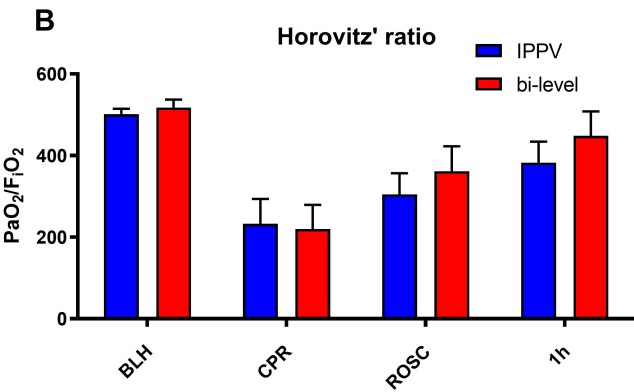

**Figure 1 Gas exchange and oxygenation values.** Gas exchange ((A) arterial partial pressure of carbon dioxide ($PaCO_2$); (B) Horovitz's ratio ($PaO_2/F_iO_2$,)) measured via blood gas analyses over different time points (baseline healthy (BLH), CPR, ROSC and 1 hour after ROSC). There were no significant differences in oxygenation or decarboxylation values during CPR and after achieving ROSC. The initial gas exchange impairment is typical and fully recovers after 1–2 h post ROSC. No major differences developed over the monitoring period afterwards.

## DISCUSSION

The present study compared standard IPPV ventilation to a novel bi-level ventilation regimen with a previously described comparable gas exchange (*Kill et al., 2014*) in a resuscitation scenario in a prospective randomized animal trial. While we could not show any significant differences in haemodynamic values or oxygenation patterns at any point during the trial, there were some results suggesting beneficial effects on overall gas exchange, lung physiology and organ perfusion.

Although there were moderate decreases in tidal volume and significantly decreased inspiratory pressures during bi-level ventilation, no significant oxygenation changes, decarboxylation impairments or decreased histological and inflammatory changes in lung tissue could be detected. Previous studies suggested mitigated pulmonary tissue damage when low-volume ventilation was applied during CPR (*Ruemmler et al., 2018*), but compared to the applied volumes in our study ($V_t$ resulting in values around 6–8 ml/kg), those tidal volumes were substantially lower ($V_t$ 2–3 ml/kg). However, we could detect a

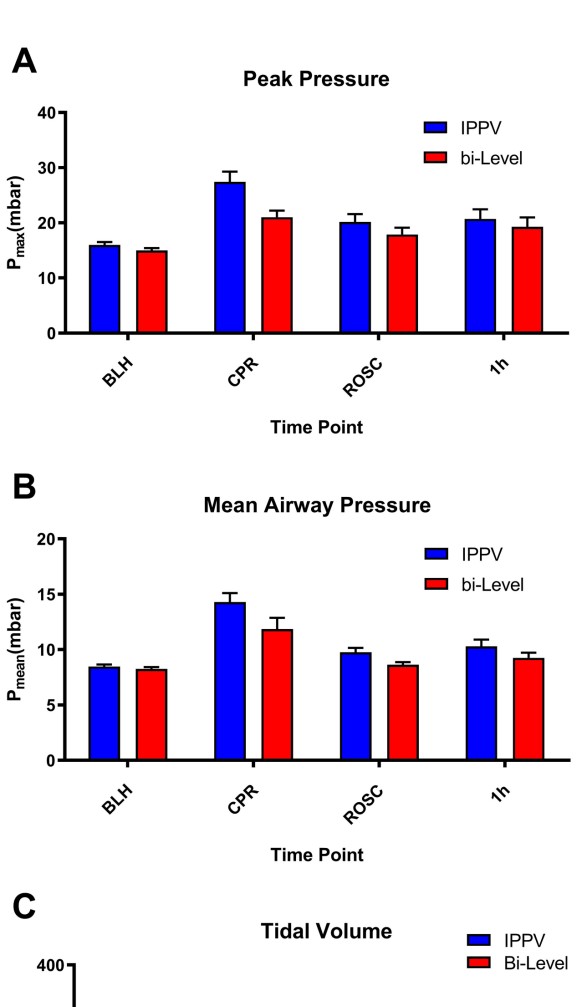

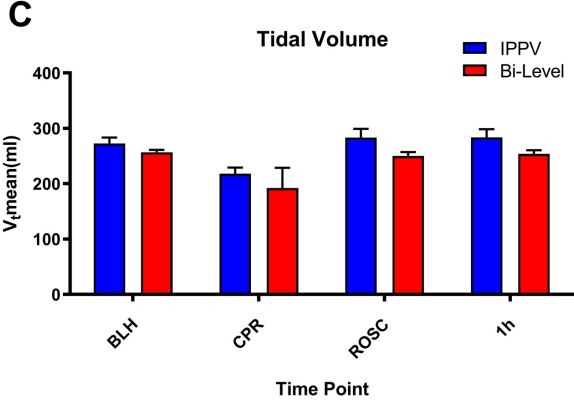

**Figure 2 Ventilatory pressures.** Ventilation parameters. Depicted are peak inspiratory pressures (A), mean airway pressures (B) and mean tidal volumes (C) over different time points (baseline healthy (BLH), CPR, ROSC and 1 hour after ROSC). Peak inspiratory pressures were significantly lower in bilevel animals ($P = 0.005$) during ROSC with no statistically significant difference in tidal volumes ($P = 0.16$) or mean airway pressures ($P = 0.11$). Post-ROSC, bilevel animals tended to lower inspiratory pressures without statistical significance. No changes were detected after the 1 h time point.

similar decrease in neuroinflammation, especially in hippocampal tissue 6 h after achieving ROSC. While those effects were small and only 4 animals survived, they were statistically significant and might point towards a better cerebral perfusion during lower volume

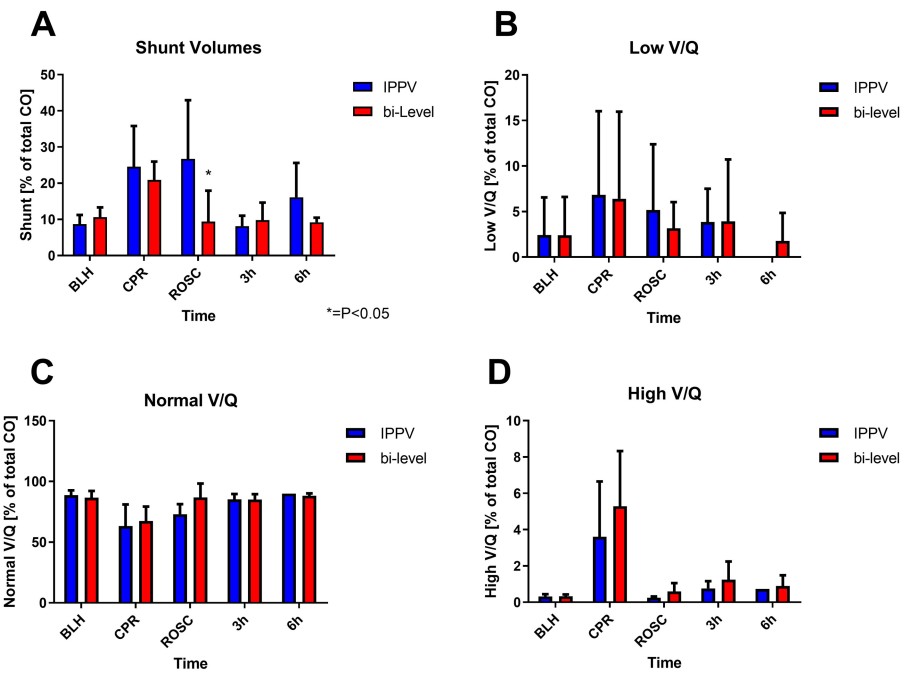

**Figure 3  Pulmonary shunt values.** MIGET measurements of pulmonary shunt- (A), low- (B), medium- (C) and high (D) ventilation/perfusion ratio volumes over different time points (baseline healthy (BLH), CPR, ROSC, 3 hours and 6 hours after ROSC). Directly after achieving ROSC, the animals showed significantly less shunt after bi-level ventilation ($P = 0.017$). There were no differences in low, middle or high ventilation/perfusion areas.

ventilation under CPR, which is consistent with prospectively generated data from our group (*Ruemmler et al., 2018*) as well as retrospective clinical analyses of out-of-hospital cardiac arrest showing favourable neurological outcome when ventilated with lower tidal volumes (*Beitler et al., 2017*).

MMIMS-MIGET measurements showed significantly less shunt volumes in bi-level ventilated animals compared to IPPV. While MIGET measurements depend on reliable cardiac output measurements and a steady state of infused inert gases, our group could repeatedly show that viable values can be obtained under extreme circumstances like CPR (*Ruemmler et al., 2018*; *Hartmann et al., 2014*). Compared to those studies, no dramatically increased high V/Q fractions indicating hyperventilation and additional shear stress could be detected. This is expected, since differences in inspiratory pressure and respiratory rates are not as pronounced as they were in these trials. The application of a constant end-expiratory pressure during ventilation might improve recruitment of compressed lung areas, thus increasing ventilated areas and allowing for lower inspiratory pressures without compromising adequate gas exchange. Additionally, lower inspiratory and consequently intrathoracic pressures might result in improved venous return and better lung perfusion, although this remains controversial (*Georgiou, Papathanassoglou & Xanthos, 2014*).

These effects combined could explain the objective differences and might even be responsible for a better pulmonary outcome when applied over prolonged periods of CPR.

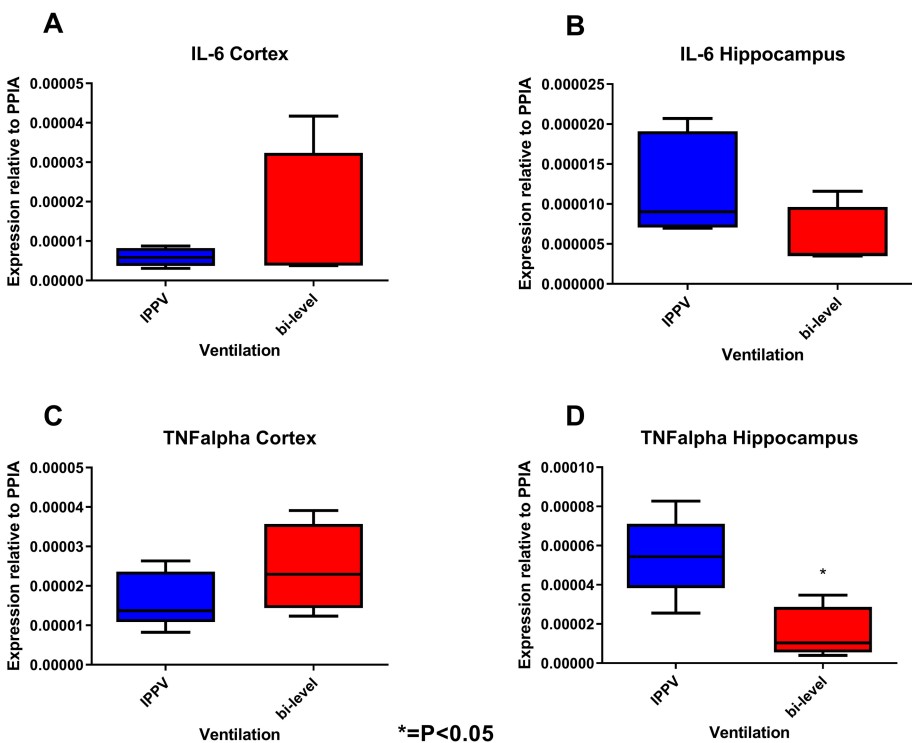

**Figure 4 Tissue sample analyses.** Inflammatory marker expressions of interleukin-6 (IL-6, A, B) and tumour necrosis factor alpha (TNF alpha, C, D) in cortical and hippocampal tissue relative to PPIA expression. While there were no statistical differences in cortical samples, hippocampal tissue showed decreased expression levels of TNFalpha and IL-6. Statistical analyses only proved to be significant for TNFalpha ($P = 0.032$).

Another explanation for an improved lung perfusion could be the use of vasopressine, which has been shown before to increase organ perfusion during resuscitation (*Meybohm et al., 2007*) and, in our own experience, can help to increase ROSC rates.

This study has several limitations. While the presented ROSC rates match with those of other resuscitation trials, no statement can be made about increased or decreased overall mortality regarding one ventilation method over the other due to restricted sample sizes, several confounding factors when working with large animals and the pilot character of the study potentially affecting CPR (*Holda et al., 2018*; *Hsu & Du, 1982*). For example, in a comparable resuscitation model, Kill et al. showed successful resuscitation rates ranging from 40–80% over several studies and with different ventilation strategies (*Kill et al., 2015*; *Kill et al., 2014*). Additionally, the no-flow time in this trial is significantly shorter compared to other resuscitation experiments and might not yield realistic results when applied to out of hospital cardiac arrest scenarios (*Kjaergaard et al., 2016*). *Tan et al. (2019)* showed a decline in ROSC rates when animals were subjected to 8 min of no-flow time compared to 4 min as presented in our study, underlining the need for further examination of the presented results.

The use of bi-level ventilation itself can be seen as controversial, since this is primarily described as a non-invasive ventilation mode, often used to support infants or patients suffering from respiratory failure (*Gillis-Haegerstr, Markstrom & Barle, 2006*). Additionally, this mode has been shown to produce increased inspiratory pressures, when applied during CPR (*Speer et al., 2017*). Although we could not observe these pressure peaks in our study and have no indication of resulting pulmonary damage, a specific pressure profile analysis might be warranted in future studies. Alternatively, the added thoracic and inspiratory pressures during CPR as well as potentially unreliable tidal volume delivery of standard ventilation modes could be counteracted with more modern, synchronized ventilation strategies like chest compression synchronized ventilation (CCSV). This has been shown to provide adequate ventilation and oxygenation (*Kill et al., 2014*) while potentially causing increased lung stress due to higher inspiratory pressures. The use of PEEP during CPR is not recommended in any resuscitation guideline to date for the lack of data and the fear of compromised circulation and cardiac filling during chest compressions. However, we could not detect any disadvantages or pressure differences during this study and showed in a previous trial, that moderate PEEP had no negative effects (*Ruemmler et al., 2018*). However, in this study, we chose to adhere to the resuscitation guidelines of the ERC, which up to this point do not recommend PEEP in their ventilation protocol during CPR. The omission of PEEP in the control group might itself be responsible for some of the effects, but since no sufficient data on this topic is available right now, separate experiments are planned by our group to explore possible benefits of different PEEP levels alone.

The direct measurement of intrathoracic pressures as a sophisticated and technically challenging method has not been considered in this study but would help to further evaluate ventilation effects and confirm potentially beneficial mechanisms and their causation. The use of MIGET measurements during CPR is sophisticated. Although several experiments of our group yielded comparable and reasonable results, validation of another experimental group in a large animal model is still missing (*Ruemmler et al., 2018*; *Hartmann et al., 2014*). Since measurements of V/Q depend on adequate cardiac output measurements and a steady state of the infused gases to eliminate, the face validity of gained results might be debatable. Apart from IL-6 and TNFalpha measurements, no cerebral parameters were taken. While it would- in theory- be feasible to measure intracranial pressure during and after CPR, we decided against inflicting additional pre-CPR trauma to the head to prevent more confounding factors from influencing inflammatory responses.

## CONCLUSION

In a porcine CPR model, bilevel ventilation was not inferior to standard IPPV and allowed for adequate gas exchange despite decreased inspiratory pressures and slightly reduced tidal volumes. In early ROSC, less pulmonary shunt could be detected, suggesting improved pulmonary ventilation/perfusion status in bilevel animals. Decreased neuroinflammatory cytokine markers point towards a better end organ perfusion under bi-level ventilation, possibly affecting neurologic outcomes. Given the short no-flow time and the character of this study, further examinations are necessary to better characterize this alternative ventilation technique and its value during resuscitation.

### Funding

The authors received no funding for this work.

### Competing Interests

The authors declare there are no competing interests.

### Author Contributions

- Robert Ruemmler conceived and designed the experiments, performed the experiments, analyzed the data, prepared figures and/or tables, authored or reviewed drafts of the paper, and approved the final draft.
- Alexander Ziebart conceived and designed the experiments, prepared figures and/or tables, and approved the final draft.
- Frances Kuropka performed the experiments, analyzed the data, prepared figures and/or tables, and approved the final draft.
- Bastian Duenges performed the experiments, analyzed the data, authored or reviewed drafts of the paper, and approved the final draft.
- Jens Kamuf conceived and designed the experiments, authored or reviewed drafts of the paper, and approved the final draft.
- Andreas Garcia-Bardon performed the experiments, prepared figures and/or tables, authored or reviewed drafts of the paper, and approved the final draft.
- Erik K. Hartmann conceived and designed the experiments, analyzed the data, prepared figures and/or tables, authored or reviewed drafts of the paper, and approved the final draft.

### Animal Ethics

The following information was supplied relating to ethical approvals (i.e., approving body and any reference numbers):

The Landesuntersuchungsamt Rheinland Pfalz provided full approval for this experiment (no.G16-1-042).

### Data Availability

Raw data are available in the Supplemental Files.

### Supplemental Information

Supplemental information for this article can be found online at http://dx.doi.org/10.7717/peerj.9072#supplemental-information.

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
