# Peer review of "Bi-Level ventilation decreases pulmonary shunt and modulates neuroinflammation in a cardiopulmonary resuscitation model"

_PeerJ, doi:10.7717/peerj.9072_

## Round 0.1 · original submission · Major Revisions

Dear authors,

In light of the reviewers' reports, I think your paper has scientific merit to be published in PeerJ. However, there are some issues which you must address before in a revised version of the text.

With respect and best regards,
Dr Palazón-Bru (academic editor for PeerJ)

·

Basic reporting

No comment

Experimental design

The research question is well defined in the introduction. However, the authors should develop the choice to compare these two particular strategies among others. The specific literature on bi-level ventilation during CPR should help to put this research question in context, especially in light of the conclusions from the study of Kill et al. (Crit Care Med. 2014 Feb;42(2):480-1.).
The investigation is rigorous and performed with high technical and ethical standard. The authors should comment on their choice to apply no PEEP during IPPV. The difference observed in this study could indeed be related to differences in end expiratory pressure level and/or in ventilation mode.
The choice of 15-17cmH2O during Bilevel should also be justified better.

Validity of the findings

The values of IL-6 and TNFa expression within the lungs, as well as histology figures should be shown, at least in supplemental material.
The measurement of IL-6 and TNFa in cortex seems to show opposite results as compared to hippocampus, even if not statistically significant. This trend could be related to a decrease in brain perfusion due to bi-level. This should be discussed in light of studies from Dr Lurie’s Lab showing that lower intrathoracic pressure during CPR is associated with better brain perfusion (Moore et al. Resuscitation. 2017 Dec;121:195-200).

Additional comments

This is a well conducted study on a very important topic. However, several end-points are not convincing, especially regarding the neuroinflammation (discrepeancies between hippocampus and cortex). The title should be reworded in order to be less categorical about the effects of bi-level ventilation on inflammation

·

Basic reporting

This is a well written and structured experimental study investigating an important part of resuscitation.
Literature is sufficient and inculdes recent studies.

Experimental design

Adaquate experimental design and relevant well defined question

Validity of the findings

Validity and interpretation of findings correct and conclusive.

Additional comments

There are some relevant points that should be clarified:

1. Methods and results:" Statistical analyses were performed using 2-way ANOVA inter-group tests with post-hoc Bonferroni correction for repeated measurements as well as Mann-Whitney U test for
single measurements with Student-Newman-Keuls post-hoc analysis via GraphPad Prism 8 software (GraphPad Software Inc., La Jolla, CA, USA). Data in the text are presented as mean (standard deviation)", but it remains unclear in the Figuers, wich test was applied as well as how the p values resulted. Please provide p values for all measuerments in Figures and tables and declare the tests (ANOVA or U-test) and explain the bar ends (SD or percentiles?).
2. Discussion: As IPPV was performed without PEEP and BILVEL with PEEP of 5, the results are not surprising me. You should discuss this limitation.
3. You cite Ref 23 as follows: "Additionally, this mode has been shown to produce excessive inspiratory pressures, when applied incorrectly during CPR(23). Although we could not observe these pressure peaks in our study and have no indication of resulting pulmonary damage, further evaluation would be necessary." What do you mean with "incorrectly applied"? How did you measure airway pressure in your study (which sampling rate)?
4. As you cite several studies dealing with CCSV you also should discuss this ventilation mode in relation to your results

If you would revise your manuscript conidering these comments I would appreciate to be allowed to review your work again.

---

## Round 0.2 · Minor Revisions

Still pending some minor modifications in your work.

·

Basic reporting

No additionnal comments

Experimental design

No additionnal comments

Validity of the findings

No additionnal comments

Additional comments

The authors addressed all my previous comments. I have no further questions.

·

Basic reporting

As metioned in my initial review, I strongly recommend to show all p-values eiteh in the graphs or in the table.

Experimental design

OK

Validity of the findings

OK

---

## Round 0.3 · accepted · Accept

All the reviewers' concerns have been correctly addressed.